# Acceptance of a Novel, Highly Palatable, Calorically Dense, and Nutritionally Complete Diet in Dogs with Benign and Malignant Tumors

**DOI:** 10.3390/vetsci10020148

**Published:** 2023-02-11

**Authors:** Reshma M. Anthony, Madison D. Amundson, John Brejda, Iveta Becvarova

**Affiliations:** 1Hill’s Pet Nutrition, Inc., Topeka 66617, KS, USA; 2Alpha Statistical Consulting, Lincoln 68502, NE, USA

**Keywords:** cancer, dogs, therapeutic food, food acceptance, eating enthusiasm, caloric intake, body weight

## Abstract

**Simple Summary:**

Dogs with cancer often lose their appetite, which leads to weight loss and affects their general health. This study investigated the acceptance of a new therapeutic, nutritionally balanced and calorically dense food by dogs with cancer. The results showed that the dogs readily ate the food, which led to improved caloric intake, increased body weight, and maintained high quality of life.

**Abstract:**

Diminished appetite and poor eating behavior accompanied by weight loss or cachexia are often reported in dogs living with cancer. This study was conducted to determine the acceptance and eating enthusiasm in dogs with cancer for a new therapeutic, nutritionally balanced, and calorically dense food designed for dogs with cancer. Adult dogs with diagnosis of cancer were recruited from general and oncology practices and were fed the study food for 28 days. Evaluations included physical examination, body weight, food intake, caloric intake, hematology and serum biochemistry, and owner assessments, namely food evaluation, quality of life, and stool scores. The dogs transitioned smoothly and tolerated the food very well. The results showed high food acceptance within the first day, with continued eating enthusiasm over the 28 days. Significant increases in food and caloric intake were observed, with the study food having a positive impact on body weight in dogs that were losing weight and helping to maintain a high quality of life. Blood laboratory parameters remained within reference ranges. Thus, the therapeutic study food was well accepted and efficacious in supporting continued eating and required caloric intake, promoting a healthy weight gain and maintaining a high quality of life in dogs with cancer.

## 1. Introduction

According to the American Veterinary Medical Association’s 2017–2018 United States dog ownership statistics, there are 79 million dogs in the United States, with roughly 39% of households owning at least one dog [1]. Advances in veterinary medicine have increased the life expectancy and quality of lives of dogs, but that also means that there is a greater chance of diagnosing cancer [2]. It has been estimated that approximately 4.2 million dogs (~53/1000 population rate) are diagnosed with cancer annually in the United States [3].

Dogs with cancer often have changes in appetite, frequently causing them to reduce food and water intake or stop eating and drinking completely [4]. This can result in complications such as weight loss and malnutrition. Cachexia, characterized by body weight loss, lean muscle mass loss, and weakness, has also been documented in dogs with cancer [5]. This could be due to the cancer itself and/or the therapies used to treat the cancer, such as chemotherapy, radiotherapy, and surgery [6]. Cancer and cancer treatments have been shown to have the potential to weaken the immune system [7,8], which may present further nutritional challenges. Appetite loss, reduced food intake, and weight loss due to the cancer or cancer therapies have been known to be the major contributors to owners’ decisions to euthanize their pets [9,10,11]. The diagnosis of cancer itself and such difficult decisions as euthanasia often take a toll on both the owner [12,13] and the attending veterinarian [14].

Reports show that most dog owners in developed countries, such as the United States and Australia, feed their dog commercial dry and wet foods [15,16]. Such commercial pet foods should comply with national/regional regulatory standards to be complete and balanced and are recommended by veterinarians following a cancer diagnosis to support the dog’s nutritional requirements [6,17]. However, upon a diagnosis of cancer in dogs, owners often become confused regarding what diets to feed to meet their dog’s nutritional needs and frequently express distrust in commercial diets [18]. They are prone to resort to other sources besides their veterinarian for dietary information such as websites, social media, or from personal experiences, which may be biased or based on erroneous information [15,16]. They also often seek alternative or complementary therapies [19,20]. Distrust in commercial pet foods also leads to feeding homemade diets [21] or raw meat-based diets [22]. Such homemade and raw diets have been shown to potentially be nutritionally incomplete [23]. Furthermore, raw diets may also contribute to a risk of foodborne illnesses from bacterial contamination, which is particularly problematic in cancer patients who are often immunocompromised due to the cancer and/or cancer therapies [23,24]. However, a report on cancer dog owners’ attitudes and practices showed that owners are willing to spend money on a commercial pet food that meets the nutritional requirements of their pets diagnosed with cancer [18].

While designing a commercial pet food, pet food manufacturers are often faced with the task of striking the right balance between nutritional attributes of the food and palatability [25]. Even the most nutritious of pet foods will be unpopular among pet owners if their pets refuse to eat it. This becomes particularly challenging because animal eating behaviors, their food preferences [26,27], and sensitivity to palatability drivers [28] are not obvious nor predictable. This can be further complicated when the animal is suffering from a disease such as cancer because the disease and the associated treatments can change eating behaviors in ways that are difficult to anticipate. Thus, a therapeutic food that meets the nutritional requirements of a dog diagnosed with cancer and is energy rich and highly palatable would be a very attractive solution.

The purpose of this study was to test the food acceptance and eating enthusiasm in dogs diagnosed with cancer of a new therapeutic food designed to be highly palatable, calorically dense, and nutrient rich. We hypothesized that canine cancer patients would have high acceptance of the new food and maintain eating enthusiasm, which would result in improved caloric and nutrient intake required to maintain quality of life of a dog living with cancer.

## 2. Materials and Methods

### 2.1. Ethics

This study was approved by the Sponsor’s Institutional Animal Care and Use Committee (CP970, 10 May 2021) and was performed in accordance with the Sponsor’s Animal Welfare Policy. Written informed consent was obtained from all owners before the commencement of the study.

### 2.2. Study Design

A single-arm, non-controlled, multicenter, prospective study was conducted in adult dogs living with a diagnosis of cancer. All dogs entering the study were fed the study food for 28 days. Evaluations included physical examinations by veterinarians to assess general health of the animals and collection of body weight, as well as to assess body fat indices (Available online: https://www.spca.org.hk/images/PDF/HillsBFIChartCanine.pdf accessed on 26 July 2022), hematology, serum biochemistry profiles, and canine pancreatic lipase (cPL), a marker for pancreatitis, at enrollment and upon study completion. Values for hematology, serum biochemistry, and cPL were compared to normal canine values to help determine overall health. See the Appendix A for body fat index scale.

Evaluations also included owner assessments through food evaluation questionnaires on food intake and food refusal, acceptance of the food, eating enthusiasm on enrollment day, and days 1–10, 14, 21 and 28 (see Appendix A for owner and veterinary surveys). For data on food intake and refusal by the participating dogs, owners weighed the amount of study food offered and eaten every day, and this was recorded in the food evaluation questionnaires. Eating enthusiasm was evaluated using a validated emoji scale for assessing food acceptability and eating enthusiasm (unpublished data). The emoji scale ranged from 1 through 7, with 1 showing the least enthusiasm to eat and 7 showing the most enthusiasm to eat (unpublished data). A score of 5, 6, or 7 on this scale indicated that the dog showed a positive food interest. Owner assessments on quality of life and stool quality (Grades 1 [liquid] to 5 [firm]) were performed through questionnaires on enrollment day and days 3, 10, 14, 21 and 28. Exit questionnaires for both pet owners and veterinarians were completed on day 28. Caloric intake of the foods that study dogs were eating before entering the study as well as the caloric intake of the study food for all dogs was calculated. Any potential relationship between adverse events and the study food was evaluated by the attending veterinarian who recorded their rationale for categorizing the adverse event as related, as applicable.

### 2.3. Study Animals

Recruitment occurred between October 2021 and June 2022. Pet owners were recruited by veterinarians from 13 general or oncology specialty practices throughout the continental United States. The goal of recruitment was to enroll dogs with a diverse group of tumor types, even those that have seemingly little influence on appetite, to evaluate nutritional intervention as a part of multimodal disease management, since appetite can be negatively affected by inflammatory process, stress, pain, nausea, dehydration, constipation, diarrhea, drugs or interference with senses [4]. This way, the population would be more representative of what is seen in clinical practice. Dogs ≥1 year of age were enrolled if they had a specific diagnosis regarding tumor type as confirmed by histopathology, including determination of whether the tumor was malignant or benign. Dogs with a presumptive diagnosis of cancer, with confirmed diagnostics pending, could also be enrolled. Dogs did not have to be undergoing treatment for their cancer. If being treated and they were receiving on-going cytotoxic chemotherapy or molecular-targeted therapy, they had to have at least 2 treatments with cytotoxic therapy that were well tolerated, with minimal impact on food intake or appetite or 2 months of treatment with molecular-targeted therapy with no impact on food intake or appetite. Dogs with a history of complete surgical removal of their tumor could be enrolled only if the cancer was considered highly malignant with a very high chance of recurrence, such as oral melanoma, osteosarcoma, or hemangiosarcoma. Dogs on non-steroidal anti-inflammatory drugs were permitted as long as they were on them for at least 10 days and dogs were not experiencing side effects. If treated with prednisone, dogs must have received it for at least 14 days at a stable dose, and they were expected to stay on the stable dose of prednisone for an additional 10 days after study enrollment. Dogs receiving surgery, chemotherapy, or radiotherapy where treatment would interfere with the first 10 days of data collection were excluded. Dogs on appetite stimulants or those that were completely anorexic for multiple days’ duration and/or had clinical evidence of dehydration and in need of intravenous fluids or immediate medical intervention were also excluded. Similarly, dogs without a definitive cancer diagnosis or those with major medical conditions such as pancreatitis, inflammatory bowel disease, chronic renal failure, or heart failure were not permitted to be enrolled. Dogs who were unable to exclusively consume the study food due to an oral tumor or advanced gingivitis, had painful, infected teeth, or had nasal tumors that would interfere with his/her sense of smell were also excluded. There were no gender specifications. Male and female, spayed and neutered dogs could be enrolled, and all breeds, both purebred and mixed breed dogs were allowed.

A dog could be removed from the study if that dog (1) experienced excessive weight loss, which was considered weight loss of >8% of initial body weight or, (2) stopped eating for 2 days or ate less than 50% of the food ration for 3 days. Overall health was assessed by the veterinarian based on a complete physical examination, hematology, serum biochemistry, and specific cPL for detecting pancreatitis, both at study entry and at study completion. All enrolled dogs lived with their owners and were allowed normal socialization and enrichment activities, both indoors and outdoors, and the conduct of the study was not to interfere with the dogs’ normal daily routine.

### 2.4. Study Food

All dogs received the study food, which was in dry form. Information about the study food, which provided 4158 Kcal metabolizable energy (ME)/kg (as fed), with 45% of calories from fat, 24% from protein, and 32% from carbohydrates, is provided in Table 1. A previous study conducted by Hill’s Pet Nutrition determined palatability by feeding the study food over 2 days to 25 healthy dogs on each day. Significantly higher intake of the study food compared to control food (*p* = 0.0116) was observed (data on file). The study food was provided in plain white bags with a color-coded label to mask the food manufacturer’s identity to study participants. All dogs included in the analysis entered the study on a variety of different foods. Supplements or treats were generally not allowed, and dog owners were instructed to maintain the dog’s feeding routine. The amount of food offered was based on the dog’s weight at enrollment, caloric content of the study foods, and input from the attending veterinarian. If the dog was not at a healthy weight at the start of the study, the amount of study food offered was based on the dog’s ideal weight. The following formula was used to calculate the dog’s daily energy requirement: 70× bodyweight kg^0.75^ with multiplier factors applied based on lifestyle factors [29]. The animals transitioned to the study food over a period of 7 days and were completely on the study food from Day 8 until the end of the study.

### 2.5. Statistical Analysis and Methods

Responses over time for intake, pet owner food evaluation questionnaire, pet quality of life assessments, and stool scores were modeled using linear and non-linear regression methods. All regression models were fit both pooled over tumor type and separately for each tumor type. Intake (grams and calories) was modeled using a quadratic-plateau model using PROC NLIN in SAS^®^, version 9.4 (SAS Institute, Cary, NC, USA; https://www.sas.com/en_us/home.html, accessed on 26 July 2022). Initial intercept and linear and quadratic slope values for the non-linear estimation procedure were determined by fitting a quadratic polynomial to the data using PROC GLIMMIX in SAS^®^, version 9.4 (SAS Institute, Cary, NC, USA; https://www.sas.com/en_us/home.html accessed on 26 July 2022).

Responses over time for pet owner food evaluation questionnaire, pet quality of life assessment and stool scores were modeled using a random coefficient model [30,31]. In a random coefficient model, both a fixed population intercept and slope as well as a random intercept and slope for each animal were estimated. The random intercepts and slopes were estimated using the RANDOM option in PROC GLIMMIX. The NOBOUND option was used to allow for a negative covariance estimate between the intercepts and slopes. Both linear and quadratic models were evaluated for each response variable. However, if the quadratic term was statistically not significant, it was dropped, and linear trends over time were chosen as the final model. The Kenward–Roger adjustment (DDFM = KR) was used to estimate the denominator degrees of freedom in the F tests [32].

Body weight and body fat index data were collected at enrollment and day 28 only. These data were analyzed using a paired *t* test to determine whether the difference between the two time points was significantly different from 0. The *t* tests were performed both pooled over tumor type and separately for each tumor type. Summary statistics for the responses from the veterinarian exit survey, pet owner exit survey, and demographic characteristics were calculated using PROC MEANS in SAS^®^, version 9.4. (SAS Institute, Cary, NC, USA; https://www.sas.com/en_us/home.html accessed on 26 July 2022).

## 3. Results

### 3.1. Population

A total of 65 dogs were enrolled, of which 25 (38.5%) had benign tumors and 40 (61.5%) had malignant tumors (Table 2). The mean age of the dogs was 9.3 years, 52.3% were female, and 60% were of mixed breed (Table 3).

### 3.2. Food Effects

In general, there were no statistically significant changes over time in owner-reported outcomes regarding how the food affected their pets (Table 4). In the few cases in which there were statistically significant changes over time in the owner’s assessment of how the foods affected their pet, the changes were very small and clinically not significant. Owner-reported strength appeared to increase slightly for dogs with benign tumors and decreased slightly for dogs with malignant tumors over the course of the study, but the trends were not statistically significant.

### 3.3. Quality of Life

Most owner-reported assessments of pet quality of life did not statistically change over time (Table 4). In the few cases in which there was a statistically significant change over time (e.g., acted like normal today, dog is joyful today for dogs with malignant tumors), the changes were very small and not clinically significant (Table 4).

### 3.4. Stool Quality

Dogs with benign tumors had consistently better stool quality scores than dogs with malignant tumors (Figure 1). Stool quality scores increased slightly over time for dogs with benign tumors and decreased slightly over time for the overall population and dogs with malignant tumors. However, none of the changes were statistically significant and consistently remained above 4 on a 5-point scale.

### 3.5. Food Intake

Four dogs were excluded from the study due to poor consumption of the diet. Intake in both grams and calories rapidly increased for all dogs, as well as for dogs with either benign or malignant tumors when the dogs were transitioned to the therapeutic food (Figure 2). For grams, the increase in intake reached a plateau at day 10 for dogs with benign tumors, day 16 for dogs with malignant tumors, and day 13 for the overall population. Similar values were observed for caloric intake, in which caloric intake increased rapidly following transition to the therapeutic food, and plateaued at day 9 for dogs with benign tumors, day 15 for dogs with malignant tumors, and day 13 for the overall population. As a result of feeding the therapeutic food, caloric intake increased by more than 300 Kcal ME/day for dogs with benign tumors, 630 Kcal ME/day for dogs with malignant tumors, and 480 Kcal ME/day for the overall population. Adjusting for metabolic body weight (MBW), caloric intake increased by almost 20 Kcal ME/day/kg MBW for dogs with benign tumors, 46 Kcal ME/day/kg MBW for dogs with malignant tumors, and 34 Kcal ME/day/kg MBW for the overall population.

### 3.6. Body Weight

Body weight increased at day 28 compared with enrollment for the overall population, those with benign tumors, and those with malignant tumors. All differences were statistically significant (Figure 3a). Statistically significant increases in body fat were seen in the overall population and in dogs with malignant tumors, and numeric increases were seen in dogs with benign tumors at day 28 compared with enrollment (Figure 3b).

### 3.7. Adverse Events and Laboratory Measures

The dogs tolerated the food well (Appendix A). The most common adverse event was hypertriglyceridemia/hypercholesterolemia/serum lipemia, which was experienced by thirteen animals, of which only one case was considered severe. There were two cases with elevated cPL and one case of hypercalcemia. Eleven dogs experienced soft stool/diarrhea/flatulence, of which all were mild or moderate, and eleven dogs had vomiting/nausea, of which two cases were considered severe. Decreased appetite was reported to be related to the study food in two dogs; no other adverse events were deemed study-food related.

Most laboratory values remained within normal ranges (data not shown); the following were the exceptions. Triglycerides were elevated at enrollment in both dogs with benign and malignant tumors (mean ± standard deviation (SD) = 167.8 ± 149.0 and 155.0 ± 130.9 mg/dL, respectively) and remained elevated at day 28 (mean ± SD = 231.4 ± 288.5 and 221.9 ± 295.4 mg/dL, respectively). Although these concentrations were above the laboratory reference range, dogs with fasting triglyceride concentrations below 1000 mg/dL are considered at low risk for developing clinical signs or for being in need of dietary intervention [33]. High-density lipoprotein cholesterol (HDL-C) increased in dogs with malignant tumors from 158.7± 28 mg/dL at enrollment to 176.0 ± 36.6 mg/dL at day 28. In dogs with benign tumors, mean HDL-C was elevated at both enrollment (185.5 ± 34.2 mg/dL) and day 28 (200.5 ± 27.4 mg/dL). Alkaline phosphatase was elevated at both enrollment (172.4 ± 322.1 U/L) and day 28 (222.1 ± 564.8 U/L) in dogs with malignant tumors.

### 3.8. Exit Interviews

Owners and the attending veterinarians were asked a series of questions regarding their perceptions on any changes in the dogs’ physical state and behavior from enrollment to day 28 (see Supplemental Material for exit surveys). Overall, both owners and veterinarians thought that the food had a positive effect on the dogs’ health and wellbeing (data not shown). Results were generally similar between dogs with benign and malignant tumors.

## 4. Discussion

The study verified the high food acceptance and continued eating enthusiasm of a new therapeutic food in dogs living with cancer. The pet owners reported that their dogs showed acceptance of the food within the first day of feeding. The mean eating enthusiasm of the study population improved within the first day compared with eating enthusiasm for the foods the dogs were being fed before entering the study. However, this improvement was not statistically significant. The reason for this is most likely that the pet owners rated the eating enthusiasm of the dogs for their current foods at the time of enrollment as generally high. The mean score on the emoji scale at the time of enrollment was 6.1, which means that the animals showed a positive interest in eating their current food. This was an expected finding because poor eating and reduced appetite often indicate reduced quality of life and are primary influencers in euthanasia decision making [11]; it is likely believed that owners would not have chosen to participate in the trial if that was the case. We also observed that the eating enthusiasm for the study food continued to stay high during the study. This study evaluated the food as a part of multimodal disease management; thus, eating enthusiasm could have been influenced by improvement in clinical status as a result of response to treatment. Nevertheless, the inclusion criteria were chosen to evaluate feeding behavior at a time when the dogs’ medication was well stabilized to minimize the effect of treatment or changes in medication on appetite.

The study population consisted of dogs with benign and malignant cancers. When the group was divided into benign and malignant subgroups, there were no differences in eating enthusiasm between groups, with the malignant group showing a significant improvement in eating enthusiasm by day 3. The eating enthusiasm continued to be high for both groups, but eating enthusiasm in the malignant group dropped toward the end of the study. This is because in several dogs with malignant cancer, the disease had progressed significantly, which was thought to have negatively influenced their eating behavior [4]. This may have contributed to the slight overall drop in eating enthusiasm for the entire population by the end of the study.

The positive changes in eating behavior exhibited by the dogs were reflected in weight gain and increases in body fat at day 28. Dogs in the study exhibited increased percentage of body fat, which is a desirable outcome for underweight cancer patients but not necessarily for those who are overweight or obese. Although this study was not designed to objectively evaluate lean muscle mass changes, the dogs experienced concurrent increases in percentages of body fat and body weight. This indicates that these patients were in positive energy balance while eating the study food, which would be expected to help minimize breakdown of muscle for energy and help to spare lean muscle mass. Results of this study underpin the importance of evaluating each patient as an individual and adjusting the amounts fed to prevent undesirable weight gain. Research in both humans and canines is limited regarding nutrient requirements, proportion of dietary macronutrients, and cancer-related weight changes and outcomes [34]. Most reports focus on management of anorexia or cancer cachexia [35,36]. It has been shown in humans that the cachexia associated with cancer is the result of alterations in basal metabolic rate and resting energy expenditure that cause detrimental changes in carbohydrate, protein, and lipid metabolism [37,38,39]. Many of these changes have also been observed in dogs with cancer [40,41]. In both humans and dogs with cancer, an association between weight loss and increased risk of death has been observed [42,43]. In one study of dogs with lymphoma or osteosarcoma, those dogs who lost even <10% of body weight experienced shorter progression-free intervals and overall survival than those who gained >10% of body weight [43]. Another study of dogs with multicentric lymphoma found that dogs who lost >5% of body weight after initial chemotherapy had significantly shorter progression-free survival and numerically shorter overall survival than dogs who maintained or gained weight [44]. Quality of life is rarely measured as an outcome in studies of dogs and cats with cancer [45]. However, in studies of humans with cancer, weight loss and cachexia have been shown to be associated with diminished quality of life [46,47]. In those few studies that have measured quality of life in companion animals, loss of appetite is one of the most frequently reported contributors to decreased quality of life [48].

A key component to the management of cancer-related cachexia in both humans and dogs is adequate nutritional and energy intake [34,49]. However, this is a challenge when dogs lose their appetite and decrease food and water intake [4]. Thus, formulating a food that is palatable and that dogs with cancer will consistently eat is an important component of care. Dogs with complete anorexia are candidates for assisted feeding nutritional support.

A limitation of this study was that it was a single-arm, non-randomized trial without a control arm, as opposed to a randomized placebo-controlled trial. However, this design was employed because of the lack of a clinically proven active control or comparator food designed to specifically meet the caloric and nutritional requirements of dogs with cancer. It was felt that it would not be ethical to feed a control food known to be less appropriate in terms of palatability and nutrition, for this population of dogs with cancer [50,51,52].

A variety of therapeutic interventions to stimulate appetite in companion animals have been used, such as mirtazapine, cyproheptadine, corticosteroids, and antiemetics. However, results have been mixed, partly because appetite stimulation is not a primary effect of these drugs and partly because there are no standards for dosing to achieve this effect [4,53]. Capromorelin, a ghrelin receptor agonist, has been shown to increase appetite in dogs and was recently approved for this indication, but was not specifically studied in dogs with cancer and is associated with side effects that may not be tolerated by this population [54]. One of the primary benefits of a food that encourages eating instead of a drug is that there should be minimal risk of adverse effects that would further compromise the fragile health status of an animal with cancer.

## 5. Conclusions

The study demonstrated high acceptance and continued eating enthusiasm of a new therapeutic food in dogs with cancer, which may have contributed to the increases in food intake and caloric intake that led to positive effects on weight in dogs that were losing weight and maintenance of a high quality of life in dogs with cancer. Thus, this study supports the use of this new therapeutic food as a favorable and nutritionally beneficial approach in the management of dogs living with cancer.

## Figures and Tables

**Figure 1 vetsci-10-00148-f001:**
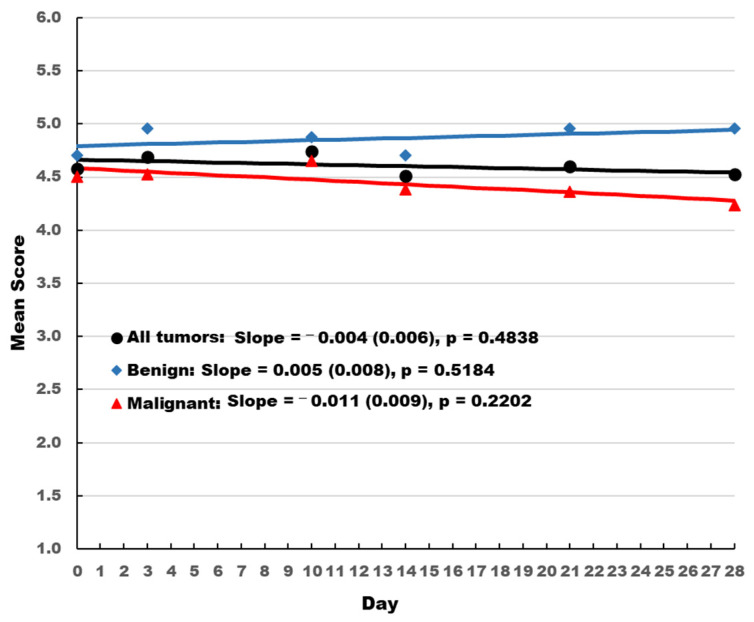
Owner-reported stool quality score. Slope and standard error is presented. The *p* values indicate whether the slope is significantly different from 0.

**Figure 2 vetsci-10-00148-f002:**
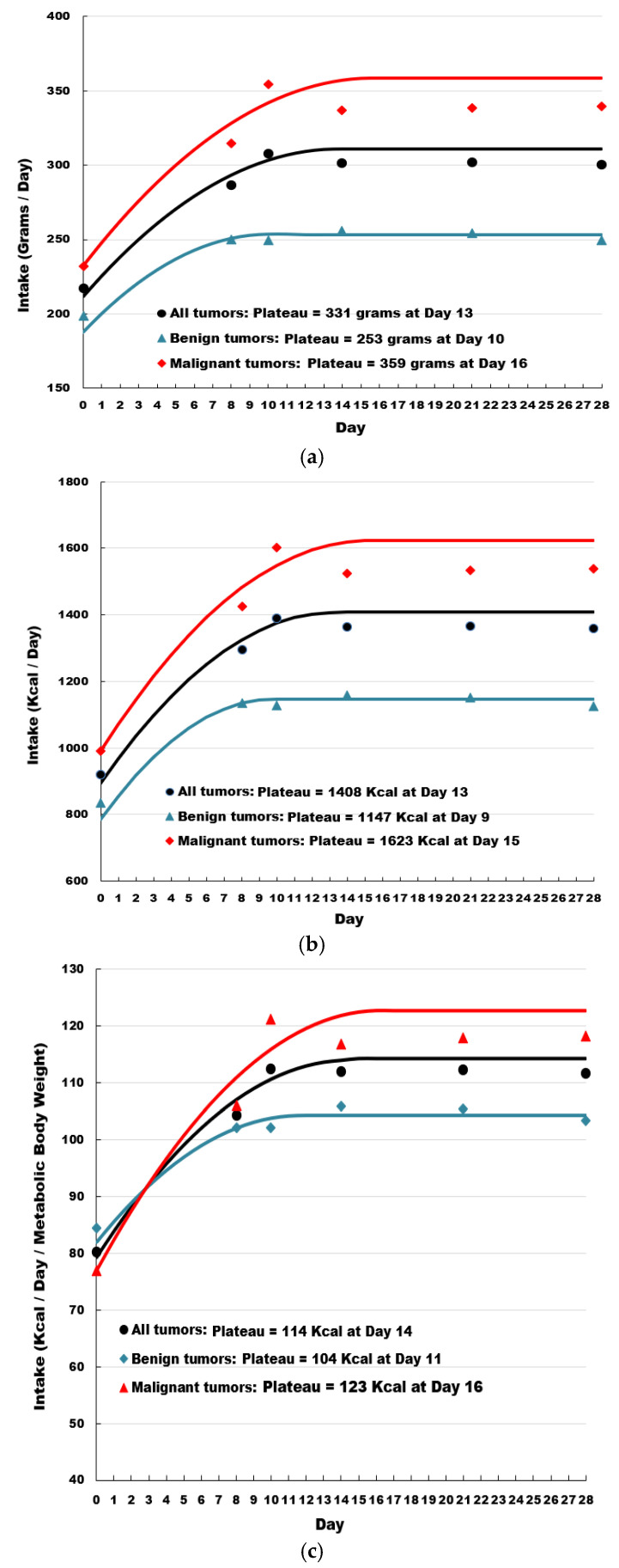
Owner-reported food intake by grams (**a**), calories (**b**), and calories per metabolic body weight (**c**).

**Figure 3 vetsci-10-00148-f003:**
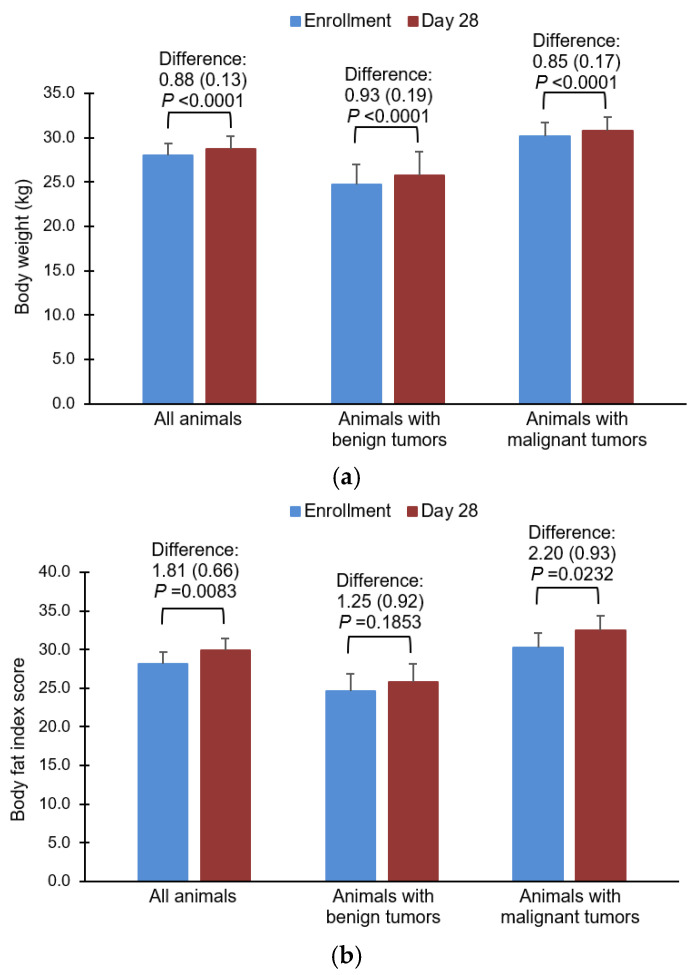
Impact of the food on body weight (**a**), and body fat index score (**b**). Data are represented as means and standard errors.

**Table 1 vetsci-10-00148-t001:** Study food ^1^ nutrient content, ingredient list, and AAFCO statement.

Nutrient	Dry Matter
Protein	29.31%
Fat	24.45%
Carbohydrate	39.29%
Total dietary fiber	8.40%
Arginine	1.50%
Carnitine	375.10 mg/kg
Total omega-3 fatty acids	1.18%
EPA	0.40%
DHA	0.26%
Total omega-6 fatty acids	3.73%
Ash	4.95%
Calcium	0.76%
Phosphorus	0.55%
Sodium	0.33%
Potassium	0.89%
Magnesium	0.09%
Vitamin E	937.00 IU/kg
Vitamin C	280.00 ppm
Vitamin A	43,200.00 IU/kg
Vitamin D	3080.00 IU/kg
**Ingredients:** Chicken, Whole Grain Wheat, Corn Gluten Meal, Chicken Fat, Hydrolyzed Chicken Liver, Soybean Meal, Brown Rice, Cracked Pearled Barley, Egg Product, Chicken Liver Flavor, Coconut Oil, Fish Oil, Ground Pecan Shells, Pork Liver Flavor, Lactic Acid, Calcium Carbonate, Potassium Citrate, Flaxseed, Dried Beet Pulp, Dried Citrus Pulp, Carrots, Iodized Salt, DL-Methionine, Pressed Cranberries, vitamins (Vitamin E Supplement, L-Ascorbyl-2-Polyphosphate (source of Vitamin C), Niacin Supplement, Thiamine Mononitrate, Calcium Pantothenate, Vitamin A Supplement, Riboflavin Supplement, Biotin, Pyridoxine Hydrochloride, Vitamin B12 Supplement, Folic Acid, Vitamin D3 Supplement), Choline Chloride, L-Tryptophan, Natural Flavors, minerals (Ferrous Sulfate, Zinc Oxide, Copper Sulfate, Manganous Oxide, Calcium Iodate, Sodium Selenite), Taurine, Mixed Tocopherols for freshness, L-Carnitine, Beta-Carotene.	
**AAFCO statement:** Animal feeding tests using AAFCO procedures substantiate that Hill’s Prescription Diet ONC Care with Chicken Dog Food provides complete and balanced nutrition for maintenance of adult dogs.	

^1^ Hill’s Prescription Diet ONC Care Canine dry. AAFCO, Association of American Feed Control Officials; DHA, docosahexaenoic acid; EPA, eicosapentaenoic acid; ppm, parts per million.

**Table 2 vetsci-10-00148-t002:** Study population by tumor type.

Tumor Type	n
Lipoma Adnexal Adenoma/cystadenoma Histiocytoma Keratinizing Lung neoplasia Extramedullary plasma cell tumor	18 2 1 1 1 1 1
Total benign	25
Mast cell tumor Malignant melanoma/melanosarcoma Perivascular wall tumor Osteosarcoma/multilobular osteochondrosarcoma Hemangiosarcoma Lymphoma Squamous cell carcinoma Mammary epithelial tumor Bronchoalveolar carcinoma Soft tissue sarcoma Splenic sarcoma Sarcoma Anal sac gland adenocarcinoma Spindle cell sarcoma Gastrointestinal stromal tumor Nasal sarcoma Prostatic	17 3 2 2 2 2 2 1 1 1 1 1 1 1 1 1 1
Total malignant	40

**Table 3 vetsci-10-00148-t003:** Patient demographics for adult dogs diagnosed with cancer. Data are represented as absolute counts or as mean ± SD.

	All Dogs	With Benign Tumor	With Malignant Tumor
**N (%)**	65 (100)	25 (38.5)	40 (61.5)
**Age (y)**	9.34 ± 2.64	8.82 ± 2.85	9.67 ± 2.48
**Weight (kg)**	28.0 ± 10.6 ^†^	24.7 ± 11.3	30.1 ± 9.7 ^1^
**Gender (n, %)**			
Female	34 (52.3)	12 (48.0)	22 (55.0)
Male	31 (47.7)	13 (52.0)	18 (45.0)
**Breed (n, %)**			
Purebreed	26 (40.0)	11 (44.0)	15 (37.5)
Mixed	39 (60.0)	14 (56.0)	25 (62.5)

^†,1^ One animal with a malignant tumor is missing enrollment body weight. Abbreviation: SD, standard deviation.

**Table 4 vetsci-10-00148-t004:** Differences from enrollment in owner-reported outcomes of food evaluation and quality of life. The *p* values in red indicate whether the slope is significantly different from 0.

Assessment	Tumor Type	Slope (SE)	95% CI Lower, Upper	*p* Value
Stool scores	All	−0.004 (0.006)	−0.017, 0.008	0.4838
	Benign	0.005 (0.008)	−0.012, 0.022	0.5184
	Malignant	−0.011 (0.009)	−0.028, 0.007	0.2202
Enthusiasm for eating food	All	−0.027 (0.008)	−0.042, −0.012	0.0008
	Benign	−0.010 (0.006)	−0.022, 0.003	0.1165
	Malignant	−0.038 (0.012)	−0.062, −0.014	0.0028
Dog’s strength	All	−0.044 (0.066)	−0.178, 0.090	0.5134
	Benign	0.032 (0.019)	−0.008, 0.072	0.1126
	Malignant	−0.098 (0.113)	−0.331, 0.135	0.3925
Dog’s energy level	All	0.050 (0.096)	−0.143, 0.244	0.6036
	Benign	0.105 (0.075)	−0.050, 0.259	0.1753
	Malignant	0.002 (0.161)	−0.329, 0.332	0.9916
Dog’s vitality	All	−0.034 (0.087)	−0.211, 0.144	0.7030
	Benign	0.063 (0.051)	−0.043, 0.168	0.2335
	Malignant	−0.107 (0.148)	−0.413, 0.199	0.4781
Vomited in last 48 h	All	0.018 (0.067)	−0.117, 0.153	0.7901
	Benign	−0.063 (0.106)	−0.282, 0.156	0.5555
	Malignant	0.079 (0.087)	−0.101, 0.259	0.3742
Dog is playful today	All	0.008 (0.099)	−0.192, 0.208	0.9351
	Benign	0.162 (0.168)	−0.185, 0.509	0.3456
	Malignant	−0.103 (0.118)	−0.349, 0.142	0.3922
Capable of doing favorite activity	All	−0.131 (0.066)	−0.265, 0.002	0.0540
	Benign	−0.009 (0.040)	−0.091, 0.073	0.8261
	Malignant	−0.225 (0.111)	−0.452, 0.003	0.0524
Acted like normal today	All	−0.200 (0.087)	−0.376, −0.024	0.0267
	Benign	−0.017 (0.061)	−0.142, 0.109	0.7864
	Malignant	−0.349 (0.148)	−0.659, −0.039	0.0293
Dog is joyful today	All	−0.131 (0.082)	−0.297, 0.036	0.1205
	Benign	0.075 (0.070)	−0.069, 0.219	0.2929
	Malignant	−0.299 (0.135)	−0.589, −0.010	0.0436
Enjoyed being near me today	All	−0.100 (0.064)	−0.249, 0.049	0.1579
	Benign	−0.022 (0.017)	−0.056, 0.012	0.1950
	Malignant	−0.181 (0.132)	−0.685, 0.322	0.2886
Dog showed normal affection	All	−0.070 (0.063)	−0.206, 0.066	0.2868
	Benign	0.018 (0.048)	−0.082, 0.117	0.7161
	Malignant	-0.122 (0.098)	−0.358, 0.115	0.2595
Enjoyed being petted or touched	All	−0.182 (0.128)	−0.446, 0.082	0.1675
	Benign	−0.004 (0.017)	−0.038, 0.031	0.8348
	Malignant	−0.309 (0.220)	−0.774, 0.156	0.1779
Dog sleeps well	All	−0.059 (0.044)	−0.148, 0.030	0.1855
	Benign	−0.015 (0.033)	−0.082, 0.052	0.6500
	Malignant	−0.107 (0.080)	−0.280, 0.066	0.2046

## Data Availability

The data are not publicly available due to confidentiality concerns.

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
