# Peer review of "Acceptance of a Novel, Highly Palatable, Calorically Dense, and Nutritionally Complete Diet in Dogs with Benign and Malignant Tumors"

_vetsci, 2023, doi:10.3390/vetsci10020148_

Round 1

Reviewer 1 Report

Lines 56-57: Reports show that most dog owners in the United States and other advanced countries feed their dog commercial dry and wet foods... – could you please specify which other countries you perceive as „advanced” and what is the main criterion to be included?

Lines 57-58: Such commercial pet foods 57 are often nutritionally complete... – Article 15 of the REGULATION (EC) No 767/2009 OF THE EUROPEAN PARLIAMENT AND OF THE COUNCIL states the type of feed is a mandatory label declaration therefore “complete feed” is a basic description of the product placed on the market.

Section 2.4. Study food Lines 167-178 – we have only your declared, general characteristics of the product used in the trial. Without further detailed data on the ingredients, chemical analysis results one cannot draw conclusions nor see the application other that a plausible manufacturer: Hill’s.

If we could return to the title in this point of the review: “calorically dense”? – data please – any comparisons? “highly palatable” – shall we refer to generally accepted protocols for palatability studies?

Reviewer 2 Report

I find it difficult to review this paper as I miss significant information about the food. I can't see anything about the energy of the food or the composition. it doesn't say how much the dogs ate of their own food before. How was that dose calculated? Did they use the same formula as you? Where they offered the same amount of their old food? How was lean body composition altered? What was the purpose of the study? I find it difficult to draw conclusions from the data you presented. I don't know anything about the food and calories offered before the trial and nothing about the food and calorie content of the food in the trial. 

Reviewer 3 Report

Thank you for a well written paper.

Comments: 

Line 56: Is 'advanced countries' a proper term? 

Inclusion criteria- could you specify which tumors were included? would a dog with a lipoma be included in this study? 

Were overweight dogs included in the study? 

How were the dogs diagnosed with cancer? how did the determination of malignancy made considering this is something you used to stratify your analysis by?  

Does excluding dogs that had a negative reaction to treatment negates the purpose of evaluating the diet in all dogs that suffer from some adverse effects to their treatment? wouldn't we want to know that the diet is well tolerated in these dogs too? 

Line 164: I assume all dogs were housed with their owners for the duration of the study, but this should be specified. 

Line 166: There should be a description of the food including ingredients, nutrients, it is not even specified that this food meet AAFCO guidelines for  adult dogs.  

Line 348: a weight gain and increased fat mass is positive in dogs that are underweight, but not in overweight and obese dogs. This should be evaluated in light of the animal's BCS

Line 367: Your statement is correct; however you did not include dogs that lost their appetite. 

Round 2

Reviewer 2 Report

Hi,

I still miss a lot of information in this study and find it difficult to review. You show that the dogs eat more calories of this food and that they increase fat index. But you don't relate it to their energy intake and you don't discuss effect of abnormal ranges of your tested parameters.

Best regards

Reviewer 3 Report

Thank you for the revision. I would suggest including more detailed information of the enrolled dogs and the type of tumor they were diagnosed with in the supplementary material. Otherwise my comments were addressed. 
